# Deep Stratigraphic Inference: A Two-Stage Training Curriculum and Heuristic Gate for High-Precision Change-Point Detection

## Abstract

Accurate geological characterization of subsurface reservoirs using well log data is essential for high-impact applications such as carbon sequestration and environmental monitoring. This task, which we term Deep Stratigraphic Inference, requires the high-precision localization of change-points within noisy time series. While transformers are powerful, a naive end-to-end regression approach fails due to training instabilities. To address this, we propose CURT-Point (Curriculum-trained Regression Transformer for Point Localization), a comprehensive framework for robust time series localization. CURT-Point's core is a Two-Stage Training Curriculum that first pre-trains the transformer as an expert classifier, then fine-tunes a specialized regression head. To maximize robustness, the framework is completed by a post-processing Hybrid System incorporating a Heuristic Gate, which achieves the best overall performance by intelligently ensembling an attention-based regression with a robust peak-finding heuristic, both derived from the same unified Transformer backbone. The effectiveness of this framework hinges significantly on two additional advancements: we show that a fusion of specific data preprocessing with an innovative constrained data augmentation tactic is crucial for dealing with real-world signal flaws, and we establish that Rotary Positional Embeddings (RoPE) play a crucial role in attaining high performance. Our final Hybrid System, validated on three real-world well-log datasets of increasing complexity, achieves state-of-the-art recall and median errors, providing a generalizable workflow for high-precision time series localization.

## 1 Introduction

Precise localization of events or change-points in intricate time series is vital across various scientific and industrial fields (Aminikhanghahi & Cook, 2017). In geoscience, it is crucial for understanding Earth's subsurface. The primary methods for examining geological formations are seismic surveys and well logging (Darling, 2005; Ellis & Singer, 2007). This study uses well-logging measurements to identify formations and map them region-wide. Well logging, which involves assessing the properties of rocks and fluids within boreholes, is essential for estimating the potential of subsurface reservoirs. The tools are deployed into boreholes to log parameters such as gamma rays, resistivity, and density (Rider, 1996; Liu et al., 2017). The aim is to construct 3D structural models of reservoirs to optimize their use and minimize ecological impact. An essential phase in model creation is well correlation, which involves aligning data from multiple wells to depict the geological landscape. We propose using formation tops, refereed to as markers, for this purpose. A fundamental principle validated by geologists and in our research is the consistent signal pattern of a specific marker within a region (Rider, 1990; Abdel Azim & Aljehani, 2022). We exploit this to perform well correlation by identifying and linking these markers across wells. The main challenge lies in interpreting the well-log data accurately. The variability in rock formations and the shortage of experts to process substantial data require advanced modeling techniques for automated identification. This is especially important for new environmental applications, such as selecting carbon capture and storage (CCS) sites, where accurate mapping is crucial for safety and efficiency (Tsuji et al., 2014; Alam et al., 2023).

Initially, we approached this as a time series classification problem to identify the different marker patterns (Katole, 2025). However, this approach proved insufficient. Some instances showed indistinct marker patterns or erroneously similar patterns appeared at incorrect depths, causing false positives. This revealed that recognizing the pattern's shape alone is insufficient; the model must also learn the global context and the expected sequence of markers. Hence, this paper introduces a sequence-to-sequence (seq2seq) method of marker propagation using transformers. Adopting a seq2seq framework allows the model to learn both the local marker signature features and the global contextual relationships between different markers. Propagating a sequence of markers together forces the model to learn the expected geological succession, providing a powerful inductive bias.

The advent of Transformers, particularly PatchTST The(Nie et al., 2023), has set a new state-of-the-art for time series representation learning. A seemingly straightforward approach for our task is to employ a multitask, end-to-end model that simultaneously classifies geological segments and regresses the precise depth of their boundaries (Li et al., 2025). However, this naive approach suffers from critical training instabilities, as the high-variance gradients from the localization task interfere with the model's ability to learn fundamental features. To overcome this, we present CURT-Point (Curriculum-trained Regression Transformer for Point Localization), a holistic and methodical investigation that addresses every stage of the modeling pipeline, from data to architecture to training. We demonstrate that a robust solution requires a synergistic combination of multiple innovations. First, we show that a data-centric foundation, combining targeted preprocessing (Hampel and Savitzky-Golay filters) (Pearson et al., 2016; Schafer, 2011) with a novel constrained data augmentation strategy, is essential for handling real-world data imperfections. Second, we conduct a rigorous architectural analysis, proving that the choice of positional embedding is important; Rotary Positional Embeddings (RoPE) (Su et al., 2024) significantly enhance performance compared to typical baselines. our core contribution is a framework we call CURT-Point (Curriculum-trained Regression Transformer for Point Localization). The centerpiece of this framework is a Two-Stage Training Curriculum, inspired by the principles of curriculum learning (Bengio et al., 2009). A naive end-to-end (E2E) approach fails because it presents the model with both easy and hard tasks simultaneously, leading to instability. We introduce a training curriculum(Bengio et al., 2009) that allows the model to first master robust feature extraction in the simpler classification task before attempting the more challenging fine-grained localization task. This involves pre-training the Transformer as a classifier to learn stable features, then freezing the backbone and fine-tuning a specialized attention-based regression head. This *specialist* model outperforms strong heuristic baselines. Finally, to address the specialist's remaining outlier errors, we introduce a Heuristic Gate, a simple but effective post-processing rule that improves robustness by correcting outlier predictions.The overall workflow of our investigation is illustrated in Figure 1.

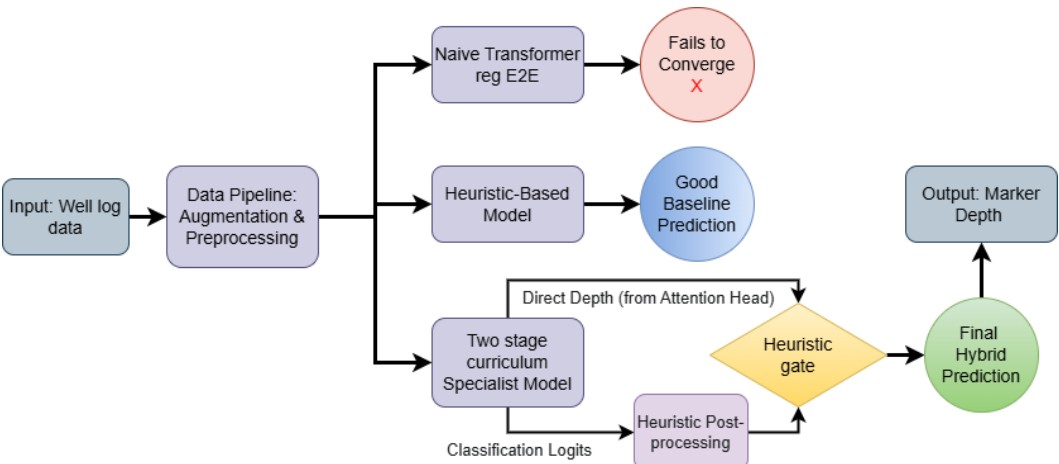

Figure 1: An overview of our experimental framework. We compare a naive baseline, a robust heuristic model, and our proposed CURT-Point system. CURT-Point uses a Heuristic Gate to integrate two distinct predictions—a high-precision direct depth and a heuristic—both derived from our core Two-Stage Specialist Model, achieving the best overall performance.

The key innovations of the CURT-Point framework are:

- A data-centric foundation featuring robust preprocessing and a novel constrained data augmentation technique to handle real-world signal imperfections.

- An optimized Transformer architecture that leverages Rotary Positional Embeddings (RoPE) to achieve monumental performance gains over standard baselines.

- A Two-Stage Training Curriculum that stabilizes the learning process, enabling the model to achieve state-of-the-art specialist precision by learning classification before fine-tuning for localization.

- A final Hybrid System inference strategy that incorporates a Heuristic Gate, which uses the model's own internal logits to correct outlier errors and achieve maximum robustness.

The paper is structured as follows. In Section 2, we review related work. Section 3 details our methodology, including the model architecture, our novel data pipeline, and the multi-stage training curriculum. Section 4 presents our comprehensive experimental results and ablation studies, and we conclude in Section 5.

## 2 RELATED WORK

Our work is based on three key areas of research. We first review the history of automated well log correlation to establish domain-specific challenges. We then discuss the state-of-the-art in Transformers for Time Series Analysis, focusing on PatchTST, before delving into the critical architectural detail of Positional Encoding in Transformers, which is a central component of our final model.

**Automated Well Log Correlation and Formation Top Detection** is a cornerstone of stratigraphic analysis in geosciences. Traditionally, this task involved manual effort, with geologists using their expertise to visually match log curves and detect distinct signal patterns (Mann & Dowell Jr, 1978). Initial computational methods aimed to automate this through signal processing. Techniques like Dynamic Time Warping (DTW) were initially popular, facilitating the alignment of log sequences via an optimal non-linear warping path (Lineman et al., 1987). Despite being useful for pairwise correlation, DTW faces challenges with multi-well correlation, as it is computationally demanding and prone to issues from signal noise and inconsistencies. Other traditional approaches include cross-correlation, statistical methods, and feature-based matching (Dashtian et al., 2011). The integration of machine learning (ML) techniques with well-log data analysis forms a robust methodology. Initially, supervised methods like Support Vector Machines (SVMs) and early neural networks utilized handcrafted features for well log classification (Al-Mudhafar, 2017). With the rise of deep learning, end-to-end methods became prominent. Recurrent Neural Networks (RNNs), such as LSTMs and GRUs, have been applied effectively to well log segmentation and marker identification (Wang et al., 2020). Furthermore, convolutional neural networks (CNNs) and hybrid models such as LSTM-CNN and LSTM-2dCNN have been used to treat well logs as 1D signals to recognize shape-based marker signatures (Imamverdiyev & Sukhostat, 2019; Salimath et al., 2025). Despite these advancements, challenges remain in managing long-range dependencies and capturing the well's global context, which transformer architectures are designed to overcome.

**Transformers for Time Series Analysis.** With its self-attention mechanism, the Transformer architecture (Vaswani et al., 2017) has become the dominant paradigm in natural language processing and is increasingly being adapted for time series analysis. Early attempts, such as the Informer and Autoformer (Zhou et al., 2021; Wu et al., 2021), focused on modifying the self-attention mechanism to handle the challenges of long-term forecasting. These models, while powerful, often treat the time series as a sequence of individual time steps, which can be computationally intensive and may not be the optimal representation for learning features. A significant breakthrough came with PatchTST(Nie et al., 2023), which we adopt as our architectural backbone. Inspired by the success of Vision Transformers (ViT) (Dosovitskiy et al., 2020), PatchTST divides a time series into patches, which are then treated as tokens. This patching mechanism has two key advantages: 1) it allows the model to learn from much longer contexts as the patch size reduces the sequence length, and 2) it helps the model learn local semantic information within each patch before the Transformer learns the global relationships between them. Furthermore, using channel-independent patching, PatchTST can be effectively applied to the multivariate signals common in well-logging data. Its

state-of-the-art performance on various time series classification (Wang et al., 2024) and forecasting benchmarks (Huang et al., 2024; Goswami et al., 2024) makes it a strong foundation for our high-precision localization task.

**Positional Encoding in Transformers.** Transformers utilize positional information to accurately model sequential dependencies (Kim et al.; Dufter et al., 2022). Initial methods employed either absolute sinusoidal embeddings or learned vectors, which struggled with sequence extrapolation. Rotary Positional Embeddings (RoPE) address this by encoding relative positions within the attention computation, enhancing translation invariance and length generalization (Su et al., 2024). RoPE has excelled in large language models like LLaMA and PaLM (Touvron et al., 2023; Chowdhery et al., 2023). Simultaneously, Attention with Linear Biases (ALiBi) integrates distance-dependent biases into attention scores, allowing models trained on short contexts to effectively handle longer sequences without retraining (Press et al., 2021). Both techniques are effective in long-sequence tasks like language modeling and time series forecasting (Deihim et al., 2023; Men et al., 2024). We conduct an ablation study on these positional embedding strategies, proposing that relative encoding, especially RoPE, is superior when pattern shape takes precedence over absolute depth.

## 3 METHODOLOGY

We introduce CURT-Point, an integrated framework characterized by a series of complementary advances. This section presents a breakdown of the framework, beginning with its mathematical formulation. Next, we describe the essential architectural elements that underpin it, the data-oriented pipeline for processing real-world signals, and lastly, the systems and training curriculums assessed.

### 3.1 PROBLEM FORMULATION

Our goal is to perform high-precision, multi-marker localization from sequential well-log data. Given a time series $X \in \mathbb{R}^{L \times C}$ representing a well log of length $L$ with $C$ channels, our goal is two-fold: 1) To produce a sequence of class labels $y_{\text{class}} \in \{0, ..., K\}^L$, and 2) To predict a set of scalar values $y_{\text{depth}} \in \{0, .., L\}^K$, representing the precise index within $L$ of the onset of each of the $K$ marker events.

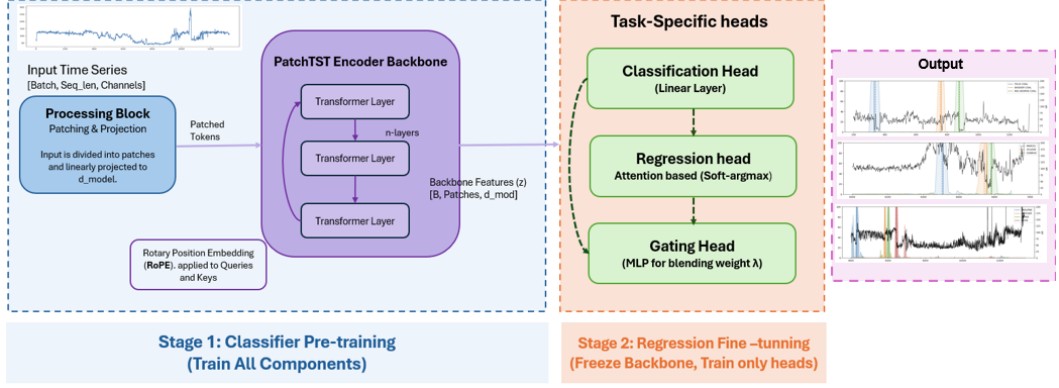

Figure 2: An overview of the CURT-Point architecture and its Two-Stage Training Curriculum. Input data is processed by a RoPE-enhanced PatchTST backbone to produce rich features. These features are then passed to a set of task-specific heads for classification and regression. The model is trained by first pre-training all components on classification (Stage 1), followed by fine-tuning the heads with a frozen backbone (Stage 2). The Output panel demonstrates the final predictions of our model on representative samples from the simple (A), moderate (B), and complex (C) datasets.

### 3.2 ARCHITECTURAL COMPONENTS

To build a model capable of learning complex patterns, our framework is constructed from a set of powerful and specialized architectural components. We begin with a state-of-the-art time series

transformer as the foundation and enhance it with carefully selected embeddings and a series of task-specific prediction heads.

We adopt the **PatchTST backbone architecture** as our core feature extractor. Its core mechanism is to divide the input time series into patches, which are then treated as tokens for a transformer encoder. This patching reduces the sequence length, allowing the model to learn from longer contexts efficiently while capturing local semantic information within each patch. To overcome the challenge of self-attention based on permutation invariance, we explored various **positional embedding** techniques. Our research indicates that relative positional embeddings are vital. The final model utilizes **Rotary Positional Embedding (RoPE) with ALiBi**, which outperforms standard embeddings by employing rotation to encode relative positions, thus improving pattern recognition independent of their absolute placement.

We designed several *specialized head* modules that attach to the backbone to perform the final prediction tasks.

- **Classification Head:** A simple linear layer that takes the backbone feature sequence and produces a logit for each class at each position.

- **Attention-Based Regression Head:** To perform precise localization, we designed a head that implements a differentiable soft-argmax. It uses the softmax of the classification logits as an attention distribution to compute a weighted average of the sequence indices, allowing the model to focus its localization decision on the most relevant parts of the sequence.

- **Gating Head:** As an advanced ablation, we designed a learnable Gating Head to find an optimal convex combination of specialist and heuristic predictions. This is a small MLP that takes the aggregated backbone features and produces a scalar blending weight $\lambda \in (0, 1)$ via a sigmoid function.

## 3.3 DATA-CENTRIC PIPELINE

Real-world well-log data require a robust pipeline to handle signal imperfections. Our framework incorporates two key data-centric stages. **Preprocessing** is applied for our most complex dataset (Dataset C). The well logs are downsampled to ensure a consistent input length, and a two-step filtering process is applied to handle noise and artifacts. We first use a Hampel filter for robust outlier removal, followed by a Savitzky-Golay filter to smooth the signal while preserving the shape of significant geological features.

To prevent the model from overfitting to the shape of a pattern while ignoring its location, we introduce a novel **constraint data enhancement** strategy. We duplicate known marker signatures and place them at incorrect *depth_idx* locations in augmented samples. The target label for these duplicated patterns is set to *no_marker*, forcing the model to learn that a pattern is only a true marker if its shape and its temporal context are correct.

## 3.4 EVALUATED SYSTEMS AND TRAINING CURRICULUMS

Simply combining the best components isn't enough; the training approach is crucial for success. Consequently, we devised and assessed multiple systems, leading to our final model that employs a tailored curriculum to address the shortcomings of naive end-to-end learning.

The **Heuristic-Based Method (*Baseline*)** uses the backbone and the classification Head. It is trained with a Cross-entropy (CE) loss only. Inference is performed via a post-processing heuristic (softmax, rolling mean, then argmax). The **Two-Stage End-to-End Model (*Specialist*)** comprises the backbone, the classification head, and the attention-based regression head. It is trained using a Two-Stage Training Curriculum, the structure of which is illustrated in relation to the model architecture in Figure 2. The curriculum begins with **Stage 1: Classifier pre-training**, where the entire model is trained solely on the classification task to build a robust and stable feature extractor. Following this, in **Stage 2: Regression Fine-tuning**, the backbone weights from the first stage are loaded and frozen. Only the head layers are then fine-tuned on our composite loss function, allowing the model to learn the high-precision localization task without corrupting the powerful, pre-trained features of the backbone.

The **Learned-Gate Model** represents a sophisticated ablation system that integrates all elements, notably the Gating Head. It is subjected to a three-phase training process, with the last phase fixing the base model parameters while exclusively training the Gating Head to merge specialist and heuristic predictions. The **hybrid system with heuristic gate** is our *final proposed solution*. It uses the fully trained Two-Stage End-to-End Model as a specialist predictor and applies a simple but powerful, rule-based gate during inference to improve robustness. The gate creates a tiered trade-off between confidence and disagreement: it overrides the specialist's prediction with the more stable heuristic prediction if the two predictions disagree significantly, with the allowable disagreement threshold increasing as the heuristic's confidence decreases. This logic allows the system to aggressively correct wildly divergent predictions while cautiously preserving the specialist's predictions when the disagreement is small. The complete logic is detailed in the appendix A.1.3.

### 3.5 TRAINING OBJECTIVE FOR THE END-TO-END MODEL

The training of our end-to-end models is guided by a composite loss function designed to address the multitask nature of the problem while enforcing known physical constraints. The total loss $L_{\text{total}}$ is a weighted sum of several components:

$$L_{\text{Spacing}} = \sum_{i=0}^{m-1}(|y_i^{m_p} - y_{i+1}^{m_p}| - |y_i^{m_t} - y_{i+1}^{m_t}|), L_{\text{Order}} = \sum_{i=0}^{m-1}\max(0, y_i^{m_p} - y_{i+1}^{m_p})$$

$$L_{\text{total}} = L_{\text{CE}} + \alpha \cdot L_{\text{L1}} + \beta \cdot L_{\text{Spacing}} + \gamma \cdot L_{\text{Order}}$$

Where $L_{\text{CE}}$ is the weighted Cross-Entropy loss for classification, $L_{\text{L1}}$ is the L1 Loss for localization precision, $L_{\text{Spacing}}$ is the Structural Consistency Penalty, and $L_{\text{Order}}$ is a crucial Order Penalty that enforces the geological constraint that markers must appear in ascending order of depth. In the equations of $L_{\text{Spacing}}$ and $L_{\text{Order}}$, $y_i^{m_p}$ refers to the predicted depth of the $i^{th}$ marker and $y_i^{m_t}$ to the true depth of the marker picked by the geologist. The specific values for the hyperparameters $\alpha$, $\beta$, and $\gamma$ were determined by ablation study, as detailed in Section 4.4.

## 4 EXPERIMENTS AND RESULTS

To validate our proposed framework, we conducted a comprehensive evaluation on three real-world gamma-ray well-log datasets. We first describe these datasets and the metrics used for the evaluation, followed by our main comparative results and a series of detailed ablation studies.

### 4.1 DATASETS

To ensure a robust and comprehensive evaluation, we curated three distinct datasets from publicly available well-log data, chosen to represent increasing levels of geological complexity. **Dataset A (Simple)**, sourced from the Colorado Basin, USA (CECMC), comprises 800 wells of 1500 ft length with high-contrast marker signatures. **Dataset B (Moderate)**, from Wyoming, USA (WOGCC), contains 500 wells of 2000 ft length with more subtle signatures and a lower signal-to-noise ratio. Our most challenging scenario, **Dataset C (Complex)**, is sourced from the Norwegian North Sea (NPD). It covers 12,000 ft per well, which we resampled to 3000 data points. This downsampling, combined with ambiguous patterns and significant noise, makes high-precision localization extremely difficult and requires the dedicated preprocessing pipeline described in Section 3.3. For all datasets, we used a standard 80/20 split for training and validation.

### 4.2 EVALUATION METRICS

To assess performance, we utilize an extensive array of metrics to encompass various facets of the localization task. Recall is employed to gauge the model's ability to recognize actual events, whereas Median Absolute Error (MedAE) is our main precision metric due to its resilience against outlier predictions commonly found in real-world data. Additionally, the Mean Absolute Error (MAE) measures the extent of these outliers and Order accuracy is used to evaluate the physical consistency of the predictions.

**Recall (%):** The percentage of markers that were correctly detected by the model within a threshold interval of 10 feet. This measures the model's sensitivity.

**Median Absolute Error (MedAE in feet):** The median of absolute differences between the predicted and true depth indices.

**Mean Absolute Error (MAE in feet):** The mean of the absolute differences. A high MAE relative to the MedAE indicates the presence of significant outlier errors, signaling a lack of robustness.

**Order Accuracy (%):** The percentage of samples with two or more markers where the predicted markers appear in the correct ascending, physically consistent order.

## 4.3 MAIN RESULTS

The main results of our investigation are summarized in Table 1. As a key baseline, the Naive End-to-End method consistently underperforms in merging tasks, illustrating the task's inherent complexity. This method involved training the entire PatchTST model, integrating its classification and attention-based regression parts, in a single phase using the loss function $[L_{loss} = L_{CE} + \alpha L_{MAE}]$. The findings underscore a trade-off between model simplicity and performance specialization across different datasets. The Heuristic-Based Method proves to be optimal for the simple dataset A with perfect results but shows significant performance deterioration with increased data complexity.

Table 1: Main results comparing all evaluated methodologies across three datasets of increasing complexity. Our final Hybrid System (Gate) achieves state-of-the-art performance, leveraging our optimal PatchTST configuration (RoPE+ALiBi) with constrained data augmentation.

| Model | Dataset A (Simple) | | | | Dataset B (Moderate) | | | | Dataset C (Complex) | | | |
|---|---|---|---|---|---|---|---|---|---|---|---|---|
| | Recall | MedAE | MAE | Order | Recall | MedAE | MAE | Order | Recall | MedAE | MAE | Order |
| DTW Baseline | 89.25 | 6.60 | 10.78 | 100 | 15.14 | 194.0 | 284.23 | 57.8 | – | – | – | – |
| LSTM-2dCNN | 97.42 | 2.0 | 2.62 | 100 | 40.37 | 76.5 | 110.21 | 100 | 8.57 | 667.98 | 801.45 | 88.2 |
| Naive End-to-End | | | | | | Fails to Converge | | | | | | |
| Heuristic-Based Method | **100** | **2.0** | **2.91** | **100** | 84.10 | 11.10 | 31.94 | 98.9 | 50.25 | 19.81 | 359.04 | 80.08 |
| Two-Stage End-to-End | 95.0 | 6.46 | 8.24 | 99 | 89.86 | 4.09 | 19.24 | 99.3 | 60.69 | 14.84 | 382.0 | 88.2 |
| Learned-Gate Model | 96.67 | 9.0 | 9.74 | 99 | 84.69 | 10.0 | 30.54 | 99.3 | 52.74 | 18.41 | **348.43** | 83.6 |
| **Hybrid System (Gate)** | 97.34 | 6.4 | 7.67 | 100 | **91.62** | **3.97** | **18.54** | **99.3** | 62.18 | 13.96 | 358.4 | **90.4** |

The Two-Stage E2E model effectively showcases its capability in moderately complex Dataset B, achieving a greater than 60% reduction in MedAE (from 11.10 to 4.09) compared to the heuristic. This advantage is more pronounced in the challenging Dataset C, where it surpasses the heuristic with a recall increase of 10 points (to 60.69%) and a 25% decrease in MedAE (to 14.84), affirming its role as a high-precision expert.

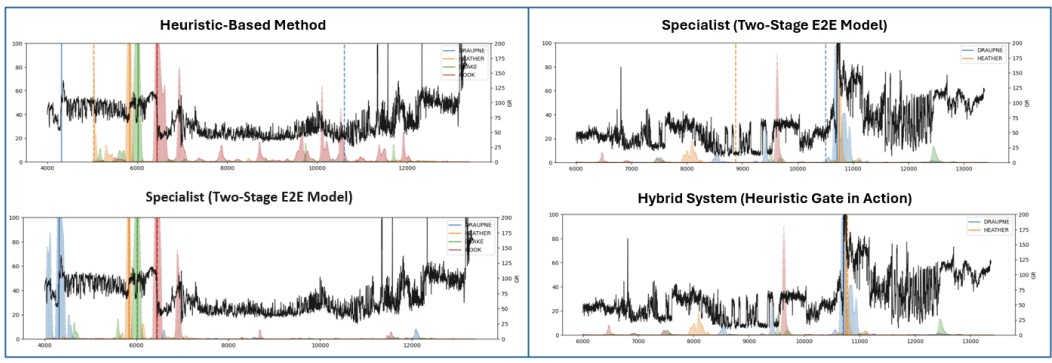

Figure 3: Qualitative results on two challenging wells from Dataset C. Ground truth is shown as solid lines, predictions as dashed lines. **(Left)** The Two-Stage E2E model demonstrates its superior precision over the Heuristic-Based Method. **(Right)** The specialist produces a significant outlier error, which our final Hybrid System's Heuristic Gate successfully identifies and corrects using the more stable prediction derived from the model's classification logits.

Although the specialist excels in precision, its high MAE indicates the presence of occasional outliers. To address this, we investigated two gating strategies. The Learned-Gate model was unsuccessful and performed worse than the specialist. This suggests that the features from the backbone,

while excellent for the primary tasks, do not contain a sufficiently clear signal for a meta-learning gate to reliably predict the specialist's own errors. In contrast, our final Hybrid System (Gate), which uses a simple, explicit rule to correct the specialist's predictions, emerges as the clear winner for non-trivial data. It achieves to maintain the precision of the specialist while significantly reducing the MAE, showcasing the best overall combination of accuracy and robustness.

## 4.4 ABLATION STUDIES

To dissect the key components of our final model and validate our design choices, we performed a series of constructive, component-wise ablation studies.

**Data-Centric Components are Foundational.** We first evaluated the impact of our data pipeline on the Heuristic–Based Method (Table 2). Our Constrained Augmentation provides a powerful and consistent benefit across all datasets, most notably nearly tripling the recall on dataset C (from 10.04% to 27.39%) and providing a 9-point recall boost on Dataset B. The targeted preprocessing pipeline, designed specifically for the resampled dataset C, provides a crucial additional boost to 30. 59% recall. These results confirm that a robust and tailored data pipeline is an essential prerequisite for any successful modeling.

Table 2: Ablation study on our data-centric components, evaluated on the Heuristic-Based Method. The Preprocessing pipeline (Hampel + SavGol filters) was only applied to Dataset C where it was necessary due to resampling. Our proposed Constrained Augmentation provides a consistent and significant benefit across all datasets.

| Configuration | Components Applied | | Dataset A (Simple) | | | Dataset B (Moderate) | | | Dataset C (Complex) | | |
|---|---|---|---|---|---|---|---|---|---|---|---|
| | Preproc. | Augment. | Recall | MedAE | MAE | Recall | MedAE | MAE | Recall | MedAE | MAE |
| 1. Baseline | (N/A) | (N/A) | 97.33 | 3.0 | 10.78 | 71.79 | 11.00 | 76.13 | 10.04 | 361.07 | 848.95 |
| 2. + Augmentation | (N/A) | ✓ | 99.33 | 4.0 | 4.49 | **81.03** | **9.00** | **34.63** | 27.39 | 404.56 | 621.20 |
| 3. + preprocessing | ✓ | ✓ | Not required | | | Not required | | | **30.59** | **30.53** | **502.66** |

**Positional Embeddings Drive Performance.** With the data pipeline established, we investigated the architectural choice of positional embeddings (Table 3). Transitioning from a typical Absolute embedding to the proposed RoPE + ALiBi combination results in an approximately 13-point recall enhancement in the Heuristic-Based Method on Dataset C. The findings demonstrate a distinct pattern: as data complexity increases, the importance of relative positional embeddings becomes crucial. We hypothesize that RoPE's superiority in this domain stems from its ability to encode relative positions via rotation. Geological patterns have a characteristic shape and relative structure that is consistent, but their absolute depth can vary significantly. By encoding position in a rotational manner, RoPE allows the self-attention mechanism to recognize these patterns in a way that is inherently invariant to shifts in their absolute location, which is a perfect inductive bias for this task. The combination of RoPE + ALiBi ultimately yields the best balance of precision and robustness across all datasets, and was therefore selected for our final models.

Table 3: Ablation on Positional Embedding Strategy, evaluated on the Heuristic-Based Method

| Positional Embedding | Dataset A | | | | Dataset B | | | | Dataset C | | | |
|---|---|---|---|---|---|---|---|---|---|---|---|---|
| | Recall | MedAE | MAE | Order | Recall | MedAE | MAE | Order | Recall | MedAE | MAE | Order |
| Absolute | 99.33 | 4.0 | 4.49 | 100 | 81.03 | 9.0 | 34.63 | 98.9 | 30.59 | 30.53 | 502.66 | 67.1 |
| ALiBi | **100** | **2.0** | **2.91** | **100** | 80.0 | 11.0 | 42.17 | 97.8 | 29.68 | 31.96 | 330.3 | 78 |
| **RoPE** | 97.66 | 3.0 | 5.58 | 100 | **85.13** | **9.0** | **30.49** | **98.9** | **43.37** | 24.35 | 407.47 | 79.4 |
| **RoPE + ALiBi** | 99.67 | 2.0 | 3.35 | 100 | 84.10 | 11.0 | 31.94 | 98.9 | 42.92 | 23.54 | 349.81 | 80.8 |

**Attention-Based Head is Superior.** Having established the optimal feature extraction setup, we evaluated the regression head architecture for our E2E models (Table 4). Our proposed Attention-Based Head is decisively superior to a standard Pooling-Based MLP baseline, confirming our hypothesis that a focused localization mechanism is critical for this task.

Table 4: Ablation on Regression Head Architecture

| Head Architecture | Two-Stage E2E | | | Learned-Gate | | | Hybrid System (Gate) | | |
|---|---|---|---|---|---|---|---|---|---|
| | Recall | MedAE | MAE | Recall | MedAE | MAE | Recall | MedAE | MAE |
| Pooling-Based | 0 | 5693.58 | 5960.51 | 0 | 5693.58 | 5960.51 | 52.42 | 18.65 | 1173.4 |
| **Attention-Based** | **58.21** | **16.17** | **367.42** | **52.74** | **18.41** | **348.43** | **57.98** | **15.96** | **305.77** |

**Loss Function Components are Critical for Precision.** Finally, with the full architecture in place, we fine-tuned the training objective (Table 5). This analysis was crucial for balancing the different learning signals from our composite loss function. The weights for the regression terms $(\alpha, \beta, \gamma)$ were selected from a range of $[1 \times 10^{-3}, 1 \times 10^{-2}]$ based on preliminary experiments. This range was chosen specifically to balance the magnitude of the different loss components. The Cross-Entropy (CE) loss typically produces values in the range of ($[0.1, 2.0]$), while the raw L1 loss, being a sum of index differences, can easily be in the hundreds. For example, a prediction off by 50 indices on two markers would yield a raw L1 loss of 100. A small $\alpha(e.g., 510^{-3})$ scales this L1 contribution down to $1000.005 = 0.5$, bringing it to a comparable scale with the CE loss and preventing the regression signal from destabilizing the model's feature learning. The analysis (Table 5) shows a clear incremental benefit from each loss component. The specialist model trained with our final loss configuration (Row 3), including the L1 ($\alpha$) and Order ($\gamma$) penalties, achieves its best precision (MedAE of 14.84). This result is then further refined by the Hybrid System's gate, which reduces the MedAE to a final best of 13.96, demonstrating that a carefully engineered loss function is essential for maximizing precision. We also investigated the addition of Spacing Penalty (Row 4), but found that it offered no consistent benefit.

Table 5: Ablation on Loss Function Components

| Configuration | $\alpha$ (L1) | $\beta$ (Spacing) | $\gamma$ (Order) | Two-Stage E2E | | | Hybrid System (Gate) | | |
|---|---|---|---|---|---|---|---|---|---|
| | | | | Recall | MedAE | Order | Recall | MedAE | Order |
| 1. CE Only | 0 | 0 | 0 | 50.25 | 19.81 | 86.3 | 50.25 | 19.81 | 86.3 |
| 2. + L1 Loss | 5e-3 | 0 | 0 | 59.7 | 14.85 | 80.8 | 61.19 | 14.59 | 83.5 |
| 3. + Order Penalty | **5e-3** | **0** | **5e-3** | **60.69** | **14.84** | **88.2** | **62.18** | **13.96** | **90.4** |
| 4. + Spacing Penalty | 5e-3 | 1e-3 | 5e-3 | 59.70 | 14.92 | 83.5 | 61.19 | 14.38 | 86.3 |
| 5. CE + Order | 0 | 0 | 1e-2 | 56.71 | 16.43 | 82.2 | 58.70 | 16.18 | 84.9 |

The ablation studies confirm our design choices. The Attention-Based head is superior to a standard pooling head (Table 4). RoPE+ALiBi provides the best performance among positional embeddings (Table 3). Our constrained augmentation is critical for improving both precision and recall (Table 2). Finally, the composite loss function with both L1 and order penalties is essential (Table 5).

## 5 CONCLUSION

In this work, we addressed the challenging task of high-precision change-point localization in geological time series. We demonstrated that naive end-to-end training of Transformers is ineffective, and proposed CURT-Point, a holistic framework for robust and accurate localization. We showed that a foundation of targeted data augmentation and RoPE encodings is critical. The framework's core is a Two-Stage Training Curriculum that produces a high-precision specialist model. To achieve maximum robustness, this specialist is integrated into a final Hybrid System that employs a simple Heuristic Gate to correct outlier predictions, achieving state-of-the-art results.

The principles of our hybrid, curriculum-based framework are highly generalizable. This methodology holds significant promise for other domains requiring high-precision event localization in noisy signals, such as seismic phase picking in geophysics, arrhythmia detection in medical ECG signals, and fault signature identification in industrial predictive maintenance.

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

# A APPENDIX

This appendix provides supplementary material to support the findings and ensure the full reproducibility of the main paper. We begin in Appendix A.1 by detailing the core implementation and hyperparameter choices, including the model architecture, the training parameters for our multi-stage curriculum, and the final settings for the Heuristic Gate. Following this, Appendix A.2 offers a deeper look into our data pipeline, with specific parameters for our preprocessing filters and a pseudocode implementation of our novel Constrained Data Augmentation. The appendix A.3 presents additional visual context, including plots of the dataset distributions, an example of the model's core task, and further qualitative results that showcase both successful predictions and characteristic failure modes of our final system. Finally, we have an Appendix A.4, on AI usage in the paper writing.

## A.1 IMPLEMENTATION DETAILS

This section provides the specific hyperparameters and implementation details required to ensure full reproducibility of our results.

### A.1.1 MODEL ARCHITECTURE HYPERPARAMETERS

The core architectural hyperparameters were determined through preliminary experiments and held constant for all reported results unless otherwise specified in an ablation study. The key parameters are detailed in Table 7.

Table 6: Key hyperparameters for the PatchTST backbone architecture.

| Hyperparameter | Value |
|---|---|
| Model Dimension (`d_model`) | 128 |
| Number of Encoder Layers (`n_layers`) | 6 |
| Number of Attention Heads (`n_heads`) | 4 |
| Patch Length (`patch_len`) | [50,100] |
| FFN Dimension (`d_ff`) | 256 |
| Dropout Rate | 0.1 |

We found that the model's performance was sensitive to both the patch length. A larger patch length of 100 was found to be optimal, significantly improving both recall and median error by allowing the model to learn from longer, more contextual patterns. Conversely, a smaller block size of 25 yielded the best results, suggesting that a finer-grained division of the sequence was beneficial.

### A.1.2 TRAINING DETAILS

All models were implemented in PyTorch using the fastai framework. We trained for a total of 12 epochs for the initial stage and 10 epochs for fine-tuning stages, using the *fit one cycle* policy with the Adam optimizer (weight decay = 0.01). The models were trained on a single NVIDIA L4 GPU.

- For Stage 1 (Classifier Pre-training), we trained for 10 epochs with a maximum learning rate of $1 \times 10^{-3}$.
- For Stage 2 (Regression Fine-tuning) and Stage 3 (Gating Head Fine-tuning), we trained for 10 epochs with a smaller maximum learning rate of $1 \times 10^{-4}$ to ensure stable convergence.
- The best bach size for Dataset A and B is 16 and for Dataset C is 4.

### A.1.3 HEURISTIC GATE PARAMETERS

The final Hybrid System uses an explicit, tiered rule-based gate to correct outliers from the Two-Stage specialist model. The gate's logic is designed to create a trade-off: it will intervene and use the safer heuristic prediction even with lower confidence if the specialist's prediction is wildly divergent, but requires very high confidence to intervene on smaller disagreements. This tiered logic is formally described in Algorithm 4.

The specific thresholds used in the algorithm were determined by a search on the validation set of Dataset C to optimize the balance between MedAE and MAE.

---

**Algorithm 2:** Tiered Heuristic Gate

---

**Input:** Heuristic Prediction $p_A$, Heuristic Confidence $\pi_A$, Specialist Prediction $p_B$
**Output:** Final Prediction $p_{\text{final}}$
1: *// Define tiered thresholds (Confidence, Max Disagreement)*
2: $T_1 \leftarrow (0.95, \text{patch-size feet})$    *// High confidence, low disagreement*
3: $T_2 \leftarrow (0.70, 500 \text{ feet})$    *// Medium confidence, medium disagreement*
4: $T_3 \leftarrow (0.30, 1500 \text{ feet})$    *// Low confidence, high disagreement*
5:
6: *// Calculate disagreement*
7: $d \leftarrow |p_A - p_B|$
8:
9: *// Apply tiered logic*
10: **if** $d > T_1[1]$ **and** $\pi_A > T_1[0]$ **then**
11:     $p_{\text{final}} \leftarrow p_A$
12: **else if** $d > T_2[1]$ **and** $\pi_A > T_2[0]$ **then**
13:     $p_{\text{final}} \leftarrow p_A$
14: **else if** $d > T_3[1]$ **and** $\pi_A > T_3[0]$ **then**
15:     $p_{\text{final}} \leftarrow p_A$
16: **else**
17:     $p_{\text{final}} \leftarrow p_B$    *// Default to trusting the specialist*
18: **end if**
19:
20: **return** $p_{\text{final}}$

---

Figure 4: Pseudocode for our proposed Tiered Heuristic Gate. The gate uses a cascade of rules to determine whether to trust the high-precision specialist prediction or fall back to the more stable heuristic prediction.

## A.2 DATA DETAILS

We provide further context on our data by providing their distribution and the input and output sequence to the transformer. **(Left) Dataset A (Colorado):** The markers are in a relatively tight and well-separated distribution. **(Center) Dataset B (Wyoming):** The distributions are wider and show more variability. **(Right) Dataset C (Norwegian North Sea):** This dataset exhibits the highest complexity. The marker depth distributions are extremely wide, spanning over 10,000 feet, and show significant overlap and multimodality, making the localization task exceptionally challenging.

Due to proprietary constraints of data extraction, we have only provided a representative snippet of Dataset C with the code submission in supplementary documents.

### A.2.1 PREPROCESSING FILTER PARAMETERS

The preprocessing pipeline for Dataset C consisted of two steps. The specific parameters, chosen to maximize signal clarity while preserving geological features, were as follows:

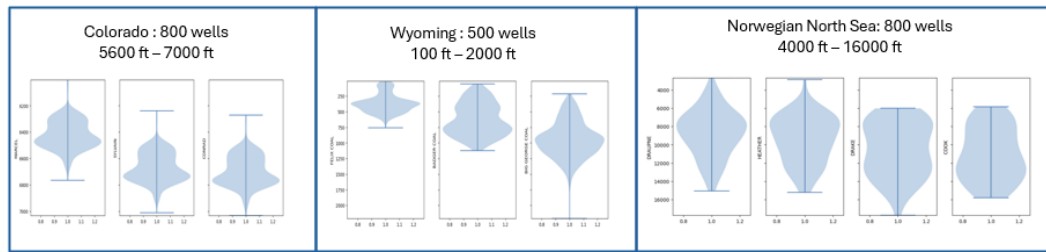

Figure 5: Violin plots showing the distribution of marker depths for each of our three datasets.

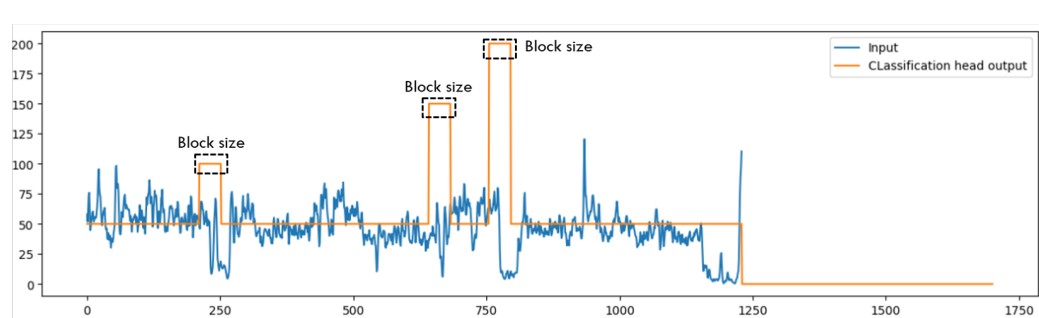

Figure 6: An example of the model's task on a sample from our dataset. The model takes the raw well-log signal (blue, "Input") and must produce a dense, point-wise classification of the geological markers (orange, "Classification head output").

Table 7: Key hyperparameters for Data processing.

| Hyperparameter | Value |
| --- | --- |
| Blocksize | 25 |
| hampel pts [mild outlier suppression] | 21 |
| hampel sigmas | 3 |
| sg pts [moderate smoothing, preserves peaks] | 21 |
| sg poly | 3 |

### A.2.2 CONSTRAINED AUGMENTATION ALGORITHM

To prevent the model from overfitting to a pattern's shape while ignoring its temporal context, we introduce a novel Constrained Data Augmentation strategy. A key challenge in our task is that signal patterns resembling true markers can appear as noise at incorrect locations. If the model learns to associate the shape of a pattern with a marker class too strongly, it will produce false positives, harming both precision and recall. Our augmentation strategy directly addresses this by teaching the model to consider context. As illustrated in Figure 7, the process involves taking a known true marker pattern, duplicating it, and pasting it at a random, incorrect location within the same time series. Crucially, the target label for this newly pasted, out-of-context pattern is set to *no_marker*. This forces the model to learn that a pattern's shape is insufficient for a positive classification; the pattern must also appear within the correct global sequence and temporal context to be considered a true marker. This process, formally described in Algorithm 8, proved essential for improving model robustness and performance across all datasets.

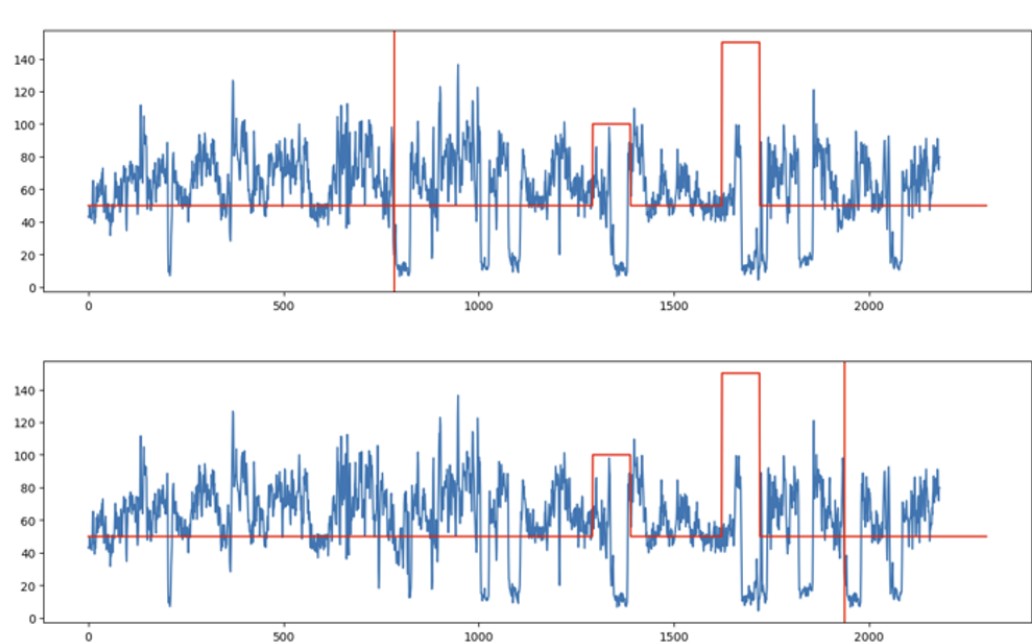

Figure 7: An example of our Constrained Data Augmentation. **(Top)** The original time series with its ground truth classification labels (red). **(Bottom)** The augmented version of the same series. A true marker pattern (e.g., the one around index 800) has been copied and pasted to a new, incorrect location (around index 1900). The target label for this pasted, out-of-context signature is explicitly set to the *no_marker* class (the flat red line), teaching the model to reject valid patterns that appear in the wrong context.

---

**Algorithm 1:** Constrained Signature Augmentation

---

**Input:** Original training set $D = \{(X_i, Y_i)\}$, Augmentation probability $p_{aug}$
**Output:** Augmented training set $D_{aug}$
1: $D_{aug} \leftarrow D$
2: **for** each sample $(X, Y)$ in $D$ **do**
3:   **if** *random() $< p_{aug}$* **then**
4:     Select a random true marker $m$ from the set of markers in $Y$
5:     Extract the signature pattern $P$ from $X$ around the location of $m$
6:     Select a random incorrect location $l_{new}$ in $X$
7:     Create a copy of the sequence, $X_{aug} \leftarrow X$
8:     Paste pattern $P$ at location $l_{new}$ in $X_{aug}$
9:     Create a copy of the labels, $Y_{aug} \leftarrow Y$
10:     Set the target label at location $l_{new}$ in $Y_{aug}$ to *no-marker*
11:     Add $(X_{aug}, Y_{aug})$ to $D_{aug}$
12:   **end if**
13: **end for**
14: **return** $D_{aug}$

---

Figure 8: Pseudocode for our proposed constrained data augmentation.

## A.3 ADDITIONAL QUALITATIVE RESULTS

We now provide additional qualitative examples on challenging samples from Dataset C.

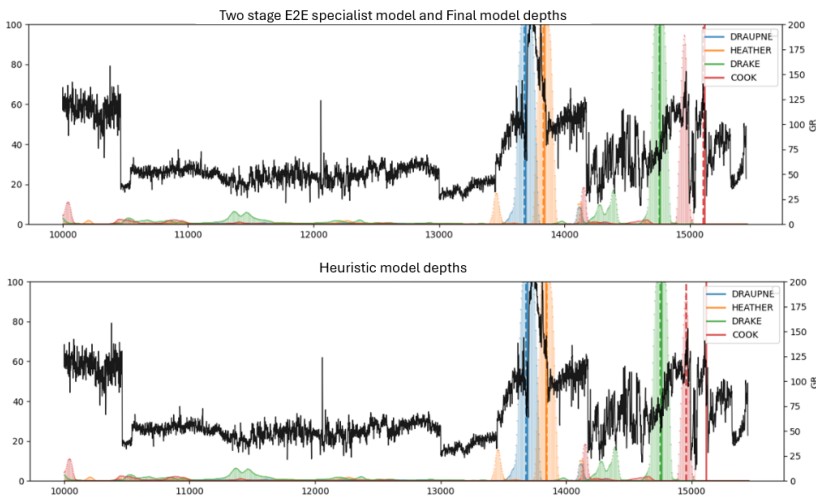

Figure 9: An additional successful prediction by our final Hybrid System on a sample from Dataset C. The model correctly identifies the marker boundary despite a noisy signal.

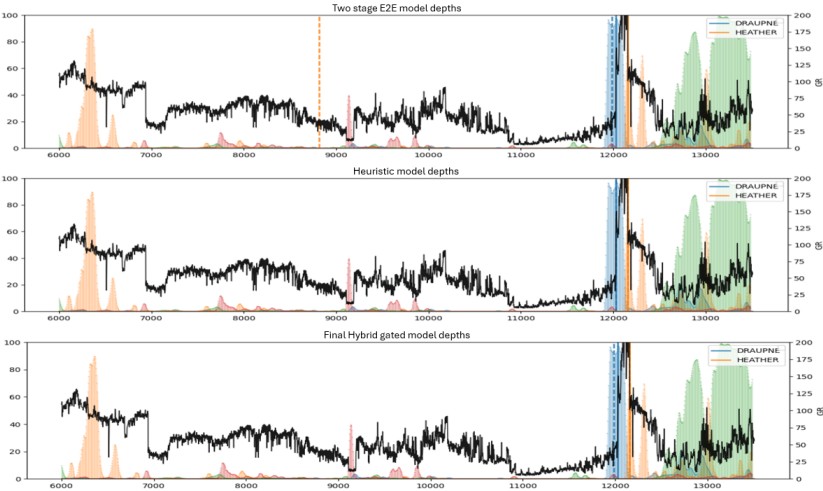

Figure 10: A case study of the Heuristic Gate in action. The Two-Stage Specialist model produces a significant outlier error (Upper), likely confused by a similar pattern earlier in the sequence. The Heuristic-Based Method provides a more stable, albeit less precise, prediction (probability distribution). Our Heuristic Gate correctly identifies the large disagreement, overrides the specialist's error, and produces a final, robust prediction (Down) that is very close to the ground truth (solid line)

## A.4 AI USAGES

The authors acknowledge the use of different AI language models (Grammarly and Google Gemini), for assistance in preparing this manuscript. Its was primarily used to improve the conciseness and structure of the text. Specific tasks included refining paragraph drafts, summarizing paragraphs, generating LaTeX code for tables and figures, alternative phrasing, and structuring section narratives.

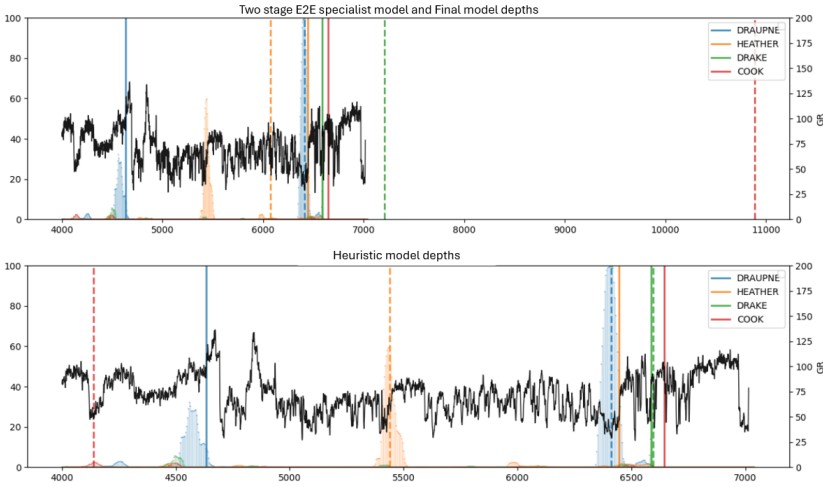

Figure 11: A characteristic failure case for our final Hybrid System. A case of unusually low sequence length, contrast, and noisy marker signature prevents the underlying classifier from producing a confident peak. As a result, both the heuristic and specialist models fail to detect the event. This highlights an area for future work to improve model sensitivity in extremely low signal-to-noise ratio conditions.

