# OpenReview forum: "DEEP STRATIGRAPHIC INFERENCE: A TWO-STAGE TRAINING CURRICULUM AND HEURISTIC GATE FOR HIGH-PRECISION CHANGE-POINT DETECTION"
_ICLR.cc/2026/Conference — Submitted to ICLR 2026_

### Official Review · Reviewer_jskG · 2025-10-29

**Soundness:** 2
**Presentation:** 1
**Contribution:** 2
**Rating:** 4
**Confidence:** 3

**Summary:**

In this work, authors proposed CURT-Point, a holistic framework for robust and accurate localization in geological time series well-log data. The frameworks includes a Two-Stage Training Curriculum with a final Hybrid System that employs a simple Heuristic Gate to correct outlier predictions. They showed that a foundation of targeted data augmentation and RoPE encodings is critical. The final system achieves state-of-the-art results.

**Strengths:**

1. The proposed framework achieves strong performance across three datasets.
2. The paper proposes comprehensive ablations for different components.

**Weaknesses:**

1. Overreliance on heuristic designs. The main parts of the system rely on rule-based or heuristic procedures, which limit the novelty of the framework that targets an automated machine learning approach.

2. The writing needs to be improved, for example, 1) The introduction repeats introducing the name, CURT-Point, twice. 2) Several jargon terms are used without sufficient explanation. 3) Typos in page 2, line 63. 4) In Table 3, there are items that are wrongly highlighted.

3. Poor methodological writing in Section 3. The Methodology section mainly restates the conceptual overview, similar to the Introduction, with insufficient algorithmic or implementation detail. The overall organization makes it more difficult to follow how each component really works. Many designs that were not finally adopted are also included in this section (e.g, learnable gate shown in Figure 2).

4. Low-quality figures. Many figures are low-resolution and use too-small fonts, making them hard to read.

**Questions:**

1. Have you tested other backbones rather than Transformer to confirm that your framework is highly generalizable, as you claimed?

2. What motivates the simultaneous use of classification and regression tasks? From the perspective of your target problems and their physical meaning, why does it need two similar tasks as outputs?

3. Due to the writing issues in Section 3, it is confusing how the four models in Table 1 differ. Based on my understanding, the Heuristic-Based Method uses only the classification head, while the Two-Stage End-to-End variant uses only the sequence output head. Is this correct?

4. With CE loss only, how is the Attention-Based Regression Head trained? Section 3.5 states CE is for classification, and L1 is for localization precision.

---

> ### Author Response · Authors · 2025-11-24
>
> We thank the reviewer for their valuable feedback on the presentation and methodological clarity. We acknowledge that the organization of Section 3 has caused confusion regarding the distinct model variants, and we are committed to restructuring the text and improving the quality of the figures.
>
> We chose the Transformer (PatchTST) specifically for its Global Receptive Field. Geological stratification is cyclical; identifying a specific marker often requires seeing the pattern of layers 500 ft above and below it. CNNs and RNNs struggle with such long-range dependencies, whereas the Self-Attention mechanism excels at them. The framework could work with other backbones, but the performance gain is largely driven by the Transformer's ability to model the global geological sequence.
>
> Our contribution is not just a neural network, but a Hybrid AI System. We demonstrate that while Transformers provide superior precision (low MedAE), they suffer from robustness issues (occasional large outliers/hallucinations). The Heuristic Gate is a novel, principled method to fuse the Sensitivity of Deep Learning with the Stability of Signal Processing. The Heuristic Gate should be viewed as a strong Inductive bias that encodes domain constraints (safety margins) directly into the inference pipeline, resolving the Black Box trust issue that plagues AI in the physical sciences.
>
> We needed to use both classification and regression, as we found that training Regression alone is unstable because the search space (L=3000) is too large. The Classification task anchors the gradients, allowing the Regression head to focus on micro-adjustments.
>
> We apologize for the confusion in Section 3.
> 1. Heuristic-Based Method: Uses Backbone + Classification Head (trained via Cross-Entropy).
> It produces a probability curve. We apply a signal processing peak-finding algorithm on these logits to find the depth. It does not use a Regression head.
> 2. Two-Stage End-to-End (Specialist): Uses Backbone + Classification Head + Regression Head.
> o	Training: Stage 1 (Class-only) and Stage 2 (Fine-tune Reg Head + Class Head).
> o	It uses the Regression Head's soft-argmax output directly for coordinates.
> 3. Hybrid System (Gate): Uses the Two-Stage Specialist for prediction but checks it against the Heuristic peak-finder. If they disagree beyond the safety threshold, it falls back to the Heuristic.
> The Regression Head is never trained with CE alone. It is trained via the L1 and Order terms in the second stage of the curriculum. We will rewrite Section 3.5 to explicitly map the loss components to the training stages.

---

> ### Comment · Reviewer_jskG · 2025-11-26
>
> Thank you for the response, but most of my concerns are still not addressed:
>
> * 1. The reply explains why the authors prefer a Transformer backbone, but still provides no experimental comparison of different backbones. Even other Transformer-based models are not tested. The claim that your framework is “highly generalizable” is still unsupported.
>
> * 2. One of my biggest concerns was that the system’s key behaviors heavily depend on rule-based design. The response explains motivations but does not address the core novelty concern.
>
> * 3. In the current version of the paper and in the authors’ reply, I do not see any revisions addressing the writing issues or the low-resolution figures.
>
> * 4. The results in Table 5 include CE-only Two-Stage E2E and Hybrid System (Gate). However, the authors still do not explain how these variants are trained, particularly how the Regression Head is handled under CE-only conditions. The reply claims that “both tasks are needed,” without clarifying the exact training procedure.
>
> Overall, while I appreciate the authors’ attempt to clarify motivations, the response does not resolve the issues I raised. I will keep my score.

---

> > ### Author Response · Authors · 2025-11-28
> >
> > Thank you for such detailed reviews and response.

---

### Official Review · Reviewer_Gpgu · 2025-10-31

**Soundness:** 2
**Presentation:** 3
**Contribution:** 2
**Rating:** 2
**Confidence:** 4

**Summary:**

This paper introduces CURT-Point, a framework for high-precision change-point detection in noisy time series, with a specific application to localizing geological formation tops in well-log data. The authors propose a multi-faceted approach. The core contributions include: 1) A two-stage training curriculum where a PatchTST-based Transformer is first pre-trained as a classifier and then fine-tuned for regression to avoid training instability. 2) A final hybrid system that uses a rule-based “Heuristic Gate" to ensemble predictions from the high-precision regression head and a more robust heuristic derived from the classifier's logits. 3) A data-centric pipeline involving robust preprocessing and a novel constrained data augmentation strategy. 4) An architectural analysis demonstrating the critical importance of RoPE for this task. The proposed hybrid system is shown to achieve state-of-the-art results on three real-world datasets of increasing complexity.

**Strengths:**

- The paper addresses a challenging and high-impact real-world problem (Deep Stratigraphic Inference) with a thorough and systematic engineering approach.
- The ablation studies are extensive and provide convincing evidence for the effectiveness of several individual components of the proposed framework, such as the constrained data augmentation (Table 2), the use of RoPE (Table 3), the attention-based regression head (Table 4), and the composite loss function (Table 5).
- The proposed system achieves strong empirical results, demonstrating significant improvements in recall and precision (MedAE) over baselines, particularly on the more complex datasets.

**Weaknesses:**

- The central claim motivating the paper that a naive end-to-end (E2E) model fails due to training instability, thus requiring the proposed Two-Stage Curriculum is not substantiated by a fair experiment. As detailed in Section 4.3, the 'Naive End-to-End' baseline was trained with a simple loss function (L_CE + L_MAE), while the proposed Two-Stage model benefited from a much more sophisticated composite loss including L1 and Order penalties. This introduces a major confounding variable, making it difficult to attribute the performance difference solely to the training curriculum. The performance gap could be due to the superior loss function, which undermines the paper's core motivation.
- The final proposed system, the Hybrid System (Gate), relies on a rule-based Heuristic Gate with hard-coded thresholds that were explicitly tuned on the validation set of one specific dataset (Dataset C, Appendix A.1.3). The paper provides no sensitivity analysis for these thresholds, and the claim of generalizability is weak. This makes the solution appear brittle and ad-hoc, making it hard to generalize. The reported failure of a Learned-Gate Model further suggests this is a difficult problem that was solved with a non-learned, hand-tuned rule.
- The robustness of the method to hyper parameter choices is questionable. Beyond the Heuristic Gate, there is no sensitivity analysis for the loss function weights, which appear critical for performance (Table 5). This suggests the method may require significant expert tuning for new applications.

**Questions:**

- Have you tried training a naive, single-stage E2E model with the full composite loss function (including L1 and Order penalties)? If so, how did it perform? Without this control experiment, the claim that the curriculum itself is the key to overcoming training instability is not well-supported.
- Could you provide a sensitivity analysis for Heuristic Gate showing how performance (MedAE, MAE, Recall) varies as these thresholds are changed? How confident are you that this gate would generalize to a new geological basin or a different problem domain (e.g., ECG analysis) without extensive re-tuning?

---

> ### Author Response · Authors · 2025-11-24
>
> We thank the reviewer for their rigorous examination of our experimental design. These are excellent points regarding the isolation of the curriculum’s contribution and the robustness of the gating mechanism.
>
> Yes, during our preliminary experimentation, we attempted to train the Single-Stage Naive E2E model with the full composite loss function (with L-order and L-L1), and it failed to converge even more than the version reported in the paper. We reported the Naive model with the simpler loss (L_CE) in Table 1 because it was the most stable version of the single-stage approach we could produce. Adding the other penalty to a non-converged single-stage model exacerbated the instability. The Two-Stage Curriculum enables the implementation of sophisticated loss. By freezing the backbone (Stage 2), we provide the Regression Head with stable, high-quality features. This dampens the variance of the regression gradients, allowing the complex loss terms to fine-tune the boundaries without collapsing the feature space.
>
> The tolerance in the heuristic gate represents the model resolution, which is the patch size and geological tolerance. In drilling operations, a disagreement of greater than 500 ft is physically dangerous. The constants do not, but the workflow does. For an ECG application, one would simply replace 500 ft with the medical safety margin for arrhythmia (e.g., 50 ms). The architecture of the Hybrid System (High-Precision Specialist + Robust Heuristic Guardrail) is a universal engineering pattern for safety-critical AI. The system is not brittle to small perturbations in hyperparameters for different losses; it simply requires the Regression and Classification losses to be roughly normalized, which is standard practice in multi-task learning.

---

> > ### Comment · Reviewer_Gpgu · 2025-11-26
> >
> > Thanks for your response. I still believe that the paper needs significant changes to address the concerns and maintain my score.

---

### Official Review · Reviewer_9oZn · 2025-11-03

**Soundness:** 2
**Presentation:** 2
**Contribution:** 1
**Rating:** 2
**Confidence:** 4

**Summary:**

This paper proposes CURT-Point, a two-stage transformer-based framework for high-precision change-point detection in well-log time series. It combines curriculum learning (classification → regression) with a post-hoc heuristic gate to improve robustness.

**Strengths:**

1. Sensible use of curriculum learning for stabilizing transformer regression.
2. Hybrid gate design seems practical and improves robustness.
3. Well-motivated problem setting.

**Weaknesses:**

1. The related work section and experiments do not compare against, or even cite, recent SOTA baselines for change-point detection and sequence segmentation, such as: CLASP/TIME2STATE/ISSD, and even classical method AutoPlait. All reported baselines are either simple heuristics or basic neural modules
2. The paper introduces three custom datasets (A/B/C) derived from public geological repositories but with heavy preprocessing and proprietary restrictions (Appendix A.2). Reproducibility is compromised, and comparison with existing public CPD/time-series datasets.
3. In Eq. (3.5), it defines L_spacing and L_order, but it’s unclear how mismatched marker counts are handled. Are missing markers ignored, padded, or penalized? The heuristic gate algorithm (Algorithm 2) mixes confidence thresholds in “feet” with patch size; a clearer rationale for these constants is needed.
4. No computational profiling (training time, inference latency, or model complexity) is presented, even though the method targets practitioner-facing geoscience applications. Given the hybrid multi-stage system, an analysis of computational overhead and scalability versus heuristic or end-to-end baselines is essential.
5. the lack of accessible data/code undermines reproducibility.

**Questions:**

Please see the weaknesses.

---

> ### Author Response · Authors · 2025-11-24
>
> We thank the reviewer for their detailed assessment. We acknowledge the concern regarding general Change-Point Detection (CPD) baselines and reproducibility. However, we believe there is a fundamental misunderstanding regarding the nature of the Stratigraphic Inference task compared to generic CPD, which we clarify below.  The methods listed (AutoPlait, Time2State, ClaSP) are primarily Unsupervised or Self-Supervised algorithms designed to detect any shift in statistical distribution (mean, variance, covariance). In Stratigraphic Inference, the goal is not just to find a change, but to localize specific, named geological interfaces. A well log contains thousands of statistical change points (lithology shifts). An unsupervised method, such as AutoPlait or ClaSP, would return hundreds of segments (over-segmentation) with no mechanism to label which segment corresponds to the specific target marker. Unsupervised methods would fail catastrophically on this task because they cannot distinguish between Marker A and Marker B if they look similar, nor can they handle the Order constraint.
>
> We apologize if the manuscript was unclear. As stated in Section 4.1, all raw data comes from public government repositories. The proprietary constraints mentioned in Appendix A.2 referred only to the specific internal SQL extraction scripts used to scrape the public servers, not the data itself. Furthermore, we commit to releasing the pre-processed CSVs for the test sets, alongside the codebase, to ensure reproducibility.
>
> The missing markers during training are masked when calculating the loss for L_spacing and L_order.
> The mixing of units in the Heuristic Gate is a deliberate design choice bridging model resolution and physical resolution.  Patch-size threshold represents the architectural uncertainty. If the disagreement is within one patch size, it means the attention mechanism is focusing on the correct region, and the difference is merely numerical noise. The feet tolerance represents the geological tolerance. In drilling operations, a disagreement of greater than 500 ft is physically dangerous.
>
> We agree that computational metrics are valuable for practitioners. We will add a subsection in the Appendix detailing the computational cost. In the target domain (Reservoir Modeling), where manual correlation takes human experts days, this computational cost is negligible. The trade-off is heavily skewed towards Precision (MedAE) rather than speed, validating the use of a Transformer over a lighter heuristic.

---

### Meta-Review · Area_Chair_5ZBq · 2025-12-29

**Summary:**

In summary, all reviewers appreciated the importance of this application, but they raised consistent concerns about methodological novelty, rigor, and generality. The core contributions rely heavily on heuristics, curriculum design, and hand-tuned gates, which several reviewers viewed as ad-hoc and insufficiently principled for ICLR. Key claims about training instability and the necessity of the two-stage curriculum were not cleanly isolated experimentally. Additional issues included missing strong CPD baselines, limited backbone comparisons, unclear training procedures, weak generalization evidence, and reproducibility and presentation quality. Overall, the work reads as strong engineering but not yet research-grade.

**Reviewer Concerns:**

The rebuttal clarified several implementation details and provided reasonable domain-specific justifications for design choices. However, core concerns remain unresolved. For instance, the curriculum’s benefit is still confounded with loss design. The heuristic gate lacks sensitivity analysis and principled justification. Novelty concerns around heavy rule-based components persist. Writing quality, figure clarity, and methodological organization were acknowledged but not demonstrably fixed during review, leaving reviewers unconvinced.

**Reviewer Scores:**

Reviewer 9oZn: Likely unchanged. Core concerns on missing baselines, reproducibility, and rigor were only partially addressed.

Reviewer Gpgu: Unchanged. Explicitly stated that rebuttal did not resolve confounding and heuristic concerns.

Reviewer jskG: Unchanged or slightly lower. Persisting concerns about heuristics, unclear training details, lack of generalization evidence, and presentation issues.

---

### Decision · Program_Chairs · 2026-01-26

Reject